# Stability-mediated epistasis constrains the evolution of an influenza protein

**Lizhi Ian Gong[1], Marc A Suchard[2], Jesse D Bloom[1,3]\***

[1]Division of Basic Sciences, Fred Hutchinson Cancer Research Center, Seattle, United States; [2]Departments of Biomathematics, Biostatistics, and Human Genetics, University of California, Los Angeles, Los Angeles, United States; [3]Computational Biology Program, Fred Hutchinson Cancer Research Center, Seattle, United States

**Abstract** John Maynard Smith compared protein evolution to the game where one word is converted into another a single letter at a time, with the constraint that all intermediates are words: WORD→WORE→GORE→GONE→GENE. In this analogy, epistasis constrains evolution, with some mutations tolerated only after the occurrence of others. To test whether epistasis similarly constrains actual protein evolution, we created all intermediates along a 39-mutation evolutionary trajectory of influenza nucleoprotein, and also introduced each mutation individually into the parent. Several mutations were deleterious to the parent despite becoming fixed during evolution without negative impact. These mutations were destabilizing, and were preceded or accompanied by stabilizing mutations that alleviated their adverse effects. The constrained mutations occurred at sites enriched in T-cell epitopes, suggesting they promote viral immune escape. Our results paint a coherent portrait of epistasis during nucleoprotein evolution, with stabilizing mutations permitting otherwise inaccessible destabilizing mutations which are sometimes of adaptive value.

## Introduction

Epistasis can play a key role in evolution, such as by constraining accessible evolutionary pathways (*Weinreich et al., 2005*; *Kryazhimskiy et al., 2011*) and increasing the role of contingency in adaptation (*Blount et al., 2008*; *Bridgham et al., 2009*). One of the simplest types of epistasis is that which occurs between mutations within a single protein. That such epistasis is common has long been considered self-evident–for example, in their seminal 1965 analysis of protein evolution, Emile Zuckerkandl and Linus Pauling wrote, "Of course … the functional effect of a given single substitution will frequently depend on the presence or absence of a number of other substitutions (*Zuckerkandl and Pauling, 1965*)." But although numerous laboratory evolution and site-directed mutagenesis experiments have demonstrated that mutations can in principle exhibit strong epistatic interactions (*Bershtein et al., 2006*; *Bloom et al., 2006*; *Lunzer et al., 2010*; *Salverda et al., 2011*), surprisingly little is known about the actual role of epistasis in natural protein evolution. A few studies have reconstructed naturally occurring mutations involved in antibiotic resistance or steroid-receptor ligand specificity (*Wang et al., 2002*; *Weinreich et al., 2006*; *Ortlund et al., 2007*; *Bridgham et al., 2009*) and found strong epistatic interactions. However, these studies have focused on small numbers of mutations pre-selected for analysis due to their putative adaptive role, and in most cases the actual temporal order of mutations is unknown.

As a result, many basic questions remain without clear answers: What is the prevalence of epistasis during protein evolution? How does epistasis arise from an evolutionary process that is conceived as proceeding through the incremental accumulation of mutations? And is it possible to coherently understand epistasis in terms of the underlying protein biophysics?

An experimental approach to address these questions is suggested by John Maynard Smith's classic analogy between protein evolution and the game where the goal is to convert one word

**\*For correspondence:** jbloom@fhcrc.org

**Competing interests:** The authors declare that no competing interests exist

**eLife digest** During evolution, the effect of one mutation on a protein can depend on whether another mutation is also present. This phenomenon is similar to the game in which one word is converted to another word, one letter at a time, subject to the rule that all the intermediate steps are also valid words: for example, the word WORD can be converted to the word GENE as follows: WORD→WORE→GORE→GONE→GENE. In this example, the D must be changed to an E before the W is changed to a G, because GORD is not a valid word.

Similarly, during the evolution of a virus, a mutation that helps the virus evade the human immune system might only be tolerated if the virus has acquired another mutation beforehand. This type of mutational interaction would constrain the evolution of the virus, since its capacity to take advantage of the second mutation depends on the first mutation having already occurred.

Gong et al. examined whether such interactions have indeed constrained evolution of the influenza virus. Between 1968 and 2007, the nucleoprotein—which acts as a scaffold for the replication of genetic material—in the human H3N2 influenza virus underwent a series of 39 mutations. To test whether all of these mutations could have been tolerated by the 1968 virus, Gong et al. introduced each one individually into the 1968 nucleoprotein. They found that several mutations greatly reduced the fitness of the 1968 virus when introduced on their own, which strongly suggests that these 'constrained mutations' became part of the virus's genetic makeup as a result of interactions with 'enabling' mutations.

The constrained mutations decreased the stability of the nucleoprotein at high temperatures, while the enabling mutations counteracted this effect. It may, therefore, be possible to identify enabling mutations based on their effects on thermal stability. Intriguingly, the constrained mutations helped the virus overcome one form of human immunity to influenza, suggesting that interactions between mutations might limit the rate at which viruses evolve to evade the immune system.

Overall, these results show that interactions among mutations constrain the evolution of the influenza nucleoprotein in a fashion that can be largely understood in terms of protein stability. If the same is true for other proteins and viruses, this work could lead to a deeper understanding of the constraints that govern evolution at the molecular level.

into another a single letter at a time passing only through intermediates that are also words (*Maynard Smith, 1970*):

$$WORD \rightarrow WORE \rightarrow GORE \rightarrow GONE \rightarrow GENE.$$

Implicit in this analogy is the idea that epistasis constrains evolution—for example, the original parent sequence does not tolerate three of the four eventual changes, as GORD, WERD and WOND are not words. We sought to similarly test for epistasis in actual protein evolution by reconstructing an extended natural evolutionary trajectory, and then also introducing each mutation individually into the original parent (*Figure 1*). While this experimental strategy is not guaranteed to find every possible epistatic interaction, it will systematically identify all mutations that have different effects in the original parent and the evolutionary intermediates in which they actually occurred. The experimental strategy in *Figure 1* also offers the possibility of determining how epistatically interacting mutations were actually fixed—for example through sequential functional intermediates as posited by Maynard Smith, or by the simultaneous or closely coupled fixation of several individually deleterious mutations (*Kimura, 1985*; *Meer et al., 2010*).

The experiment outlined in *Figure 1* requires a protein for which it is possible both to reconstruct the natural evolution and to assay for the functions that contribute to biological fitness. Human H3N2 influenza A virus is exceptionally suited to the first requirement, as the extensive availability of contemporary and historical sequences enables the detailed reconstruction of evolutionary trajectories. We focused on the 498-residue nucleoprotein (NP). Although NP's evolution is less rapid and medically infamous than that of its surface counterparts hemagglutinin and neuraminidase, NP still accumulates roughly one amino-acid substitution per year (*Rambaut et al., 2008*). Crucially for our experiment,

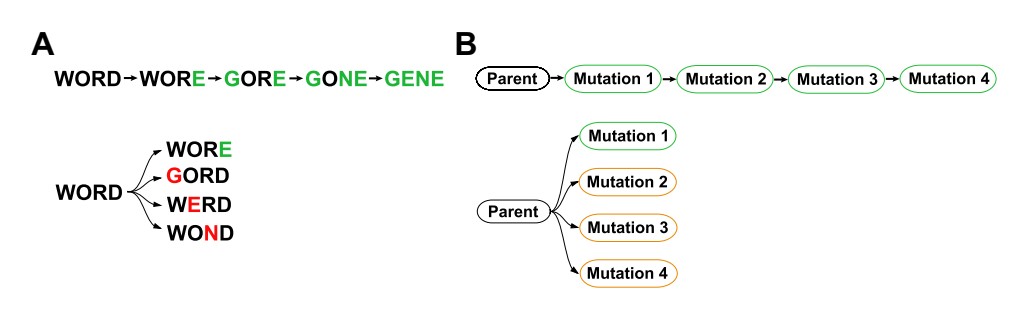

**Figure 1**. Outline of experiment designed to parallel Maynard Smith's analogy. The actual evolutionary trajectory involves the accumulation of mutations, and consists of a series of evolutionary intermediates. We recreate and experimentally assay each of these evolutionary intermediates. We also introduce each mutation individually into the original parent sequence, and experimentally assay these single mutants. If Maynard Smith is correct, each of the naturally occurring evolutionary intermediates should be a functional protein. However, some of the single mutants could exhibit impaired function if there is significant epistasis among mutations along the evolutionary trajectory.

NP's primary function—serving as a scaffold for viral RNA during transcription and genome packaging (*Portela and Digard, 2002*; *Ye et al., 2006*)—occurs within the interior of infected cells, and so is probably fairly authentically represented in tissue-culture assays.

NP is also a target of cytotoxic T lymphocytes (CTLs), and so is under continual pressure for change in CTL epitopes (*Berkhoff et al., 2004*, *2007*; *Valkenburg et al., 2011*)—a pressure partially countered by the fact that some of these epitopes are under functional constraint (*Rimmelzwaan, et al., 2004a*; *Berkhoff et al., 2005*, *2006*). CTL selection in influenza is thought to be weaker than antibody selection on the viral surface proteins, and so much of NP's evolution is shaped by stochastic forces such as population bottlenecks and hitchhiking with antigenic mutations in the surface proteins (*Rambaut et al., 2008*; *Bhatt et al., 2011*)—stochastic forces that in some cases can also accelerate the fixation CTL escape mutations (*Gog et al., 2003*). Our experiments do not measure these complexities of immune pressure as they do not include CTL selection, but as described later in this paper, existing data enable us to identify adaptive CTL-escape mutations.

In the work reported here, we use the strategy in *Figure 1* to synthesize information about influenza's natural evolution with our own experiments to examine epistasis in NP evolution. We find that epistasis constrains both the sequence evolution and ongoing adaptation of NP, and that the mechanistic basis for most of this epistasis can be understood in remarkably simple terms.

## Results

### Several mutations are under strong epistatic constraint

We focused on the evolutionary trajectory separating NPs from two human H3N2 strains isolated 39 years apart, A/Aichi/2/1968 and A/Brisbane/10/2007 (*Figure 2*). To map this trajectory, we developed a probabilistic technique to estimate the posterior distribution of mutational events (*Minin and Suchard, 2008*; *O'Brien et al., 2009*) and, original to this work, their time-orderings along an unknown phylogenetic tree within the BEAST software package (*Drummond et al., 2012*). Each sample from this posterior distribution represents a mutational path from Aichi/1968 to Brisbane/2007, which in turn can be represented as a directed graph through protein sequence space (*Figure 2—figure supplement 1*). Summarizing these graph samples effectively integrates over uncertainty in the tree and substitution process, and yields the marginal posterior distribution of the evolutionary trajectory from Aichi/1968 to Brisbane/2007 (*Figure 2*). The most probable trajectory consists of 39 mutational steps at 33 sites (5 mutations revert; 1 site mutates to two identities). The fact that NP sequences are available for every year since 1968 allows us to reconstruct the trajectory with remarkable precision: there are >$10^{46}$ possible orderings of 39 mutations, yet we can confidently identify the sequences of 25 of the evolutionary intermediates; the remainder fall along regions of the trajectory where two or more mutations occurred in an unknown order (*Figure 2*).

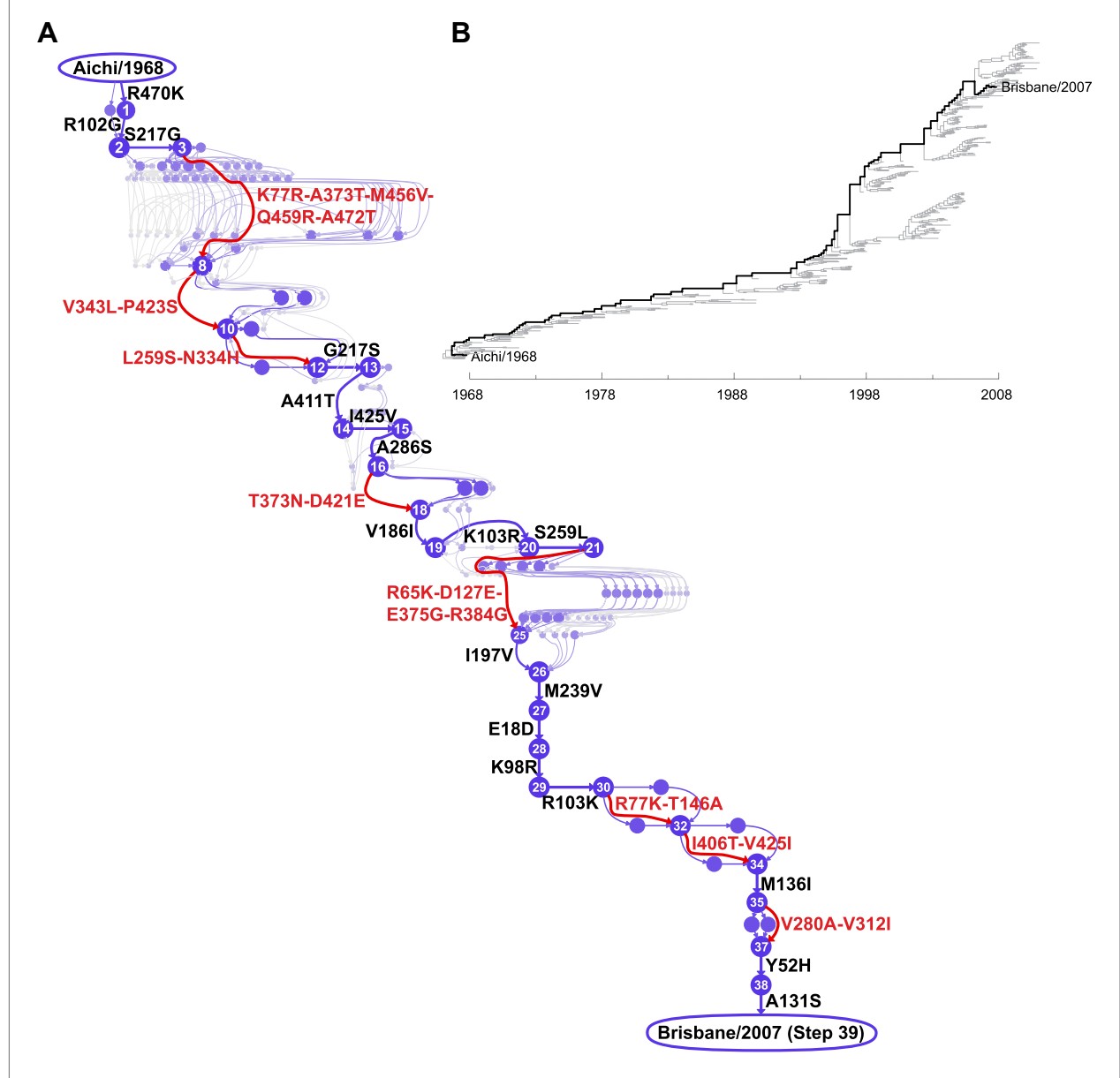

Figure 2. Inferred evolutionary trajectory. (A) Evolutionary trajectory through protein sequence space from Aichi/1968 to Brisbane/2007 NP. Each circle represents a unique inferred sequence, with areas and intensities proportional to the probability that sequence was part of the evolutionary trajectory (*Figure 2—figure supplement 1*). Mutations for which the parent and descendent are clearly resolved are in black; mutations that occurred in an unknown order are in red. High-confidence evolutionary intermediates have numeric labels. The estimated date of occurrence of each mutation is shown in *Figure 2—figure supplement 2*. (B) Phylogenetic tree of the human H3N2 NPs from 1968 to 2011 that were used to infer the evolutionary trajectory. The lines of descent connecting Aichi/1968 and Brisbane/2007 to their common ancestor are in black. Data and computer code are provided in *Figure 2—source code 1*.

The following source code and figure supplements are available for figure 2:

**Source code 1**. The sequence data and source code used to generate the evolutionary trajectory and phylogenetic tree.

**Figure supplement 1**. Construction of the evolutionary trajectory.

**Figure supplement 2**. Dates at which mutations fixed along the evolutionary trajectory in *Figure 2A*.

We created plasmids encoding each of the high-confidence protein intermediates along the trajectory, using the same codons found in the natural sequences but not introducing any of the synonymous mutations that occurred during this timeframe. We used a mini-replicon system to assess the transcriptional activity of each NP in combination with polymerase proteins (PB2, PB1, PA) from the human H3N2 strain A/Nanchang/933/1995 (*Figure 3—figure supplement 1*). All evolutionary intermediates exhibited high activity (*Figure 3A*), supporting Maynard Smith's notion that evolution proceeds through functional sequences.

We then introduced each mutation individually into the parent Aichi/1968 NP and measured its effect on activity (*Figure 3B*). Most single mutants were highly functional, but three (L259S, R384G, V280A) exhibited large decreases in activity. These three mutations are also deleterious in the background of polymerase proteins from Aichi/1968 and Brisbane/2007 (*Figure 3—figure supplement 3*), suggesting that the deleterious effect is intrinsic to NP itself.

Because RNA transcription is essential for influenza replication, impaired activity should be devastating to viral fitness. To confirm this, we used reverse genetics (*Hoffmann et al., 2000*) to generate GFP-carrying viruses (*Bloom et al., 2010*) with the polymerase genes from Nanchang/1995 and the remaining genes from the lab-adapted A/WSN/1933 (H1N1) strain (*Figure 3—figure supplement 2*). Viruses with the parent (Aichi/1968) or final (Brisbane/2007) NP grew to comparably high titers in tissue culture (*Figure 3C*), but growth of the three transcriptionally impaired mutants was dramatically lower (>1000-fold lower for L259S and R384G, and >20-fold lower for V280A). However, we observed good growth of the first high-confidence evolutionary intermediates in which these mutations were actually fixed (*Figure 3C*).

## Effects of the epistatically constrained mutations in the evolutionary intermediates in which they occurred

To understand how the three individually deleterious mutations fixed along the line of descent without a substantial fitness cost, we examined the evolutionary trajectory. L259S fixed in an unknown order with another mutation (N334H) in an evolutionary intermediate that we have labeled Step 10 (*Figure 4A*). L259S is deleterious to activity and growth in both Aichi/1968 and Step 10, while N334H has no major effect in either background (*Figure 4A*). But N334H rescues the deleterious effect of L259S in both Aichi/1968 and Step 10, indicating that fixation of L259S was enabled by N334H. We cannot determine the order of these two mutations during the evolution of the virus: they could have occurred simultaneously, or one could have preceded the other by a sufficiently small amount of time that no influenza isolates were sequenced in the intervening time period.

The next individually deleterious mutation, R384G, fixed in Step 21 in an unknown order with several other mutations (*Figure 4B*). Immediately prior to this, the individually deleterious mutation L259S reverted in the transition from Step 20 to Step 21. R384G is deleterious to activity and growth in both Aichi/1968 and Step 20 (*Figure 4B*). However in Step 21, the deleterious effect on activity disappears while that on growth diminishes, indicating that reversion of L259S alleviates the impact of R384G. The further addition of E375G to Step 21 containing R384G eliminates the remaining growth impairment— but E375G alone fails to rescue R384G in the background of Aichi/1968 (*Figure 4B*). Note that E375G has also been previously reported to partially compensate for R384G in slightly different genetic backgrounds (*Rimmelzwaan et al., 2004a*). Fixation of R384G was therefore partially enabled by the preceding reversion of L259S, with further assistance from E375G. S259L clearly preceded R384G, but we are unable to resolve whether the second enabling mutation (E375G) occurred before, after, or simultaneously with R384G.

The final individually deleterious mutation, V280A, fixed in Step 35. Although V280A is deleterious in Aichi/1968, it has no negative impact in Step 35 (*Figure 4C*). M136I, which immediately preceded V280A in the natural evolution, mostly rescues its deleterious effect in the background of Aichi/1968 (*Figure 4C*). Therefore, V280A was enabled by mutations (including M136I) that occurred prior to its own fixation.

## Most of the mutational effects are mediated by protein stability

What is the mechanistic explanation for these mutational effects? None of the identified epistatically interacting residues are in contact in the monomeric or known oligomeric NP crystal structures (*Ye et al., 2006, 2012; Ng et al., 2008*), nor are any of them in the protein's RNA-binding groove (*Figure 5* and *Figure 5—figure supplement 1*). We therefore hypothesized that the individually deleterious

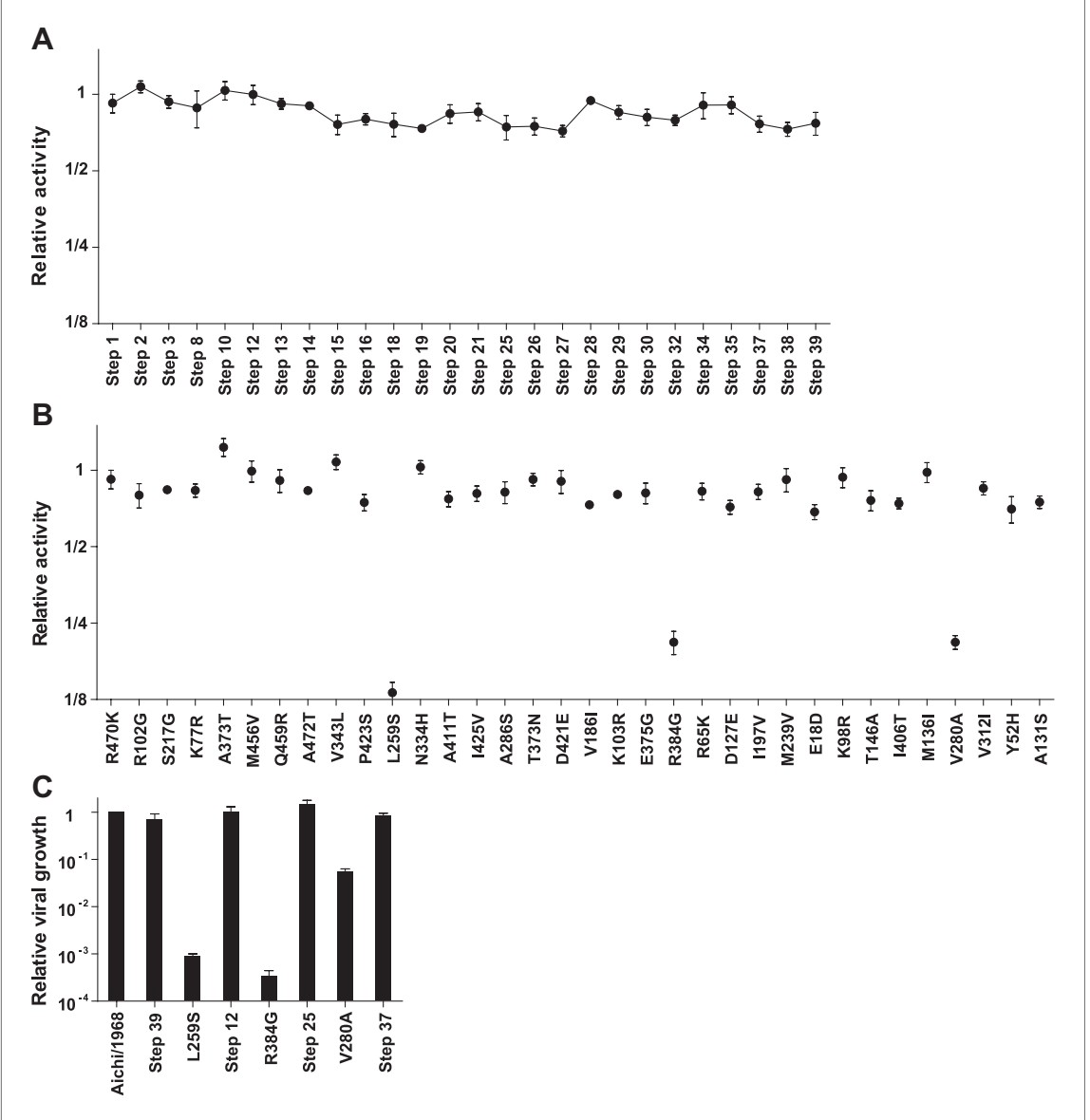

**Figure 3**. Three mutations are strongly deleterious when introduced individually into the parent Aichi/1968 NP, despite eventually becoming fixed along the evolutionary trajectory without apparent negative effect. (**A**) The transcriptional activity for the high-confidence evolutionary intermediates, quantified using a GFP reporter (**Figure 3—figure supplement 1**). Activity is scaled so that the parent Aichi/1968 NP has an activity of one. The numeric labels of the evolutionary intermediates match those used in **Figure 2A**. (**B**) The change in activity caused by introducing each mutation individually into the parent Aichi/1968 NP. The deleterious effects on activity caused by L259S, R348G, and V280A are not caused by the genetic background of influenza polymerase proteins, as these three mutants are impaired regardless of whether the polymerase proteins are derived from Nanchang/1995, Aichi/1968, or Brisbane/2007 (**Figure 3—figure supplement 3**). (**C**) All three of the individual mutations that reduce activity also impair growth, yet there is no defect in the growth of viruses carrying the NPs of the first high-confidence evolutionary intermediates in which these individually deleterious mutations were actually fixed. Viral growth is quantified as described in **Figure 3—figure supplement 2**. Numerical data are in **Figure 3—source data 1–3**.

The following source data and figure supplements are available for figure 3:

**Source data 1**. Summary of transcriptional activity data (mean and standard error) for all variants from this study.

**Source data 2**. Summary of viral growth data (mean and standard error) for all variants for which this property was measured in this study.

*Figure 3. Continued on next page*

*Figure 3. Continued*

**Source data 3**. Transcriptional activity data in the alternative polymerase genetic backgrounds shown in *Figure 3—figure supplement 3*.

**Figure supplement 1**. Schematic of NP transcriptional activity assay.

**Figure supplement 2**. Schematic of viral growth assay.

**Figure supplement 3**. Effects of key NP mutations on transcriptional activity in different polymerase genetic background of the viral polymerase genes.

mutations might destabilize NP, and that the epistasis might be mediated by counterbalancing stabilizing mutations. This hypothesis is consistent with the fact that N334H rescues L259S, and reversion of L259S in turn partially rescues R384G, despite the lack of physical contact among these residues.

We began by pairing N334H with each of the individually deleterious mutations in the background of the parent Aichi/1968 NP. N334H rescues activity for each of these mutations (*Figure 6A*). N334H also rescues growth for L259S and V280A, and largely rescues growth for R384G (*Figure 6B*). To test if this rescue was related to in vivo protein levels, we quantified NP in transfected human cells (*Figure 6C,D*). The parent Aichi/1968 NP and its N334H mutant were present at comparably high levels, but levels were markedly reduced for variants carrying each of the three individually deleterious mutations. Addition of N334H to these mutants restored wild-type protein levels, indicating that N334H can counteract the decrease in protein levels associated with the individually deleterious mutations.

To see if these changes in in vivo protein levels correlated with global protein stability, we purified the NP variants with an additional mutation in the tail loop (*Ye et al., 2006*) that enabled us to obtain monomeric RNA-free protein that exhibited the expected alpha-helical circular-dichroism spectrum (*Figure 6—figure supplement 2*). All NP variants exhibited similar circular-dichroism spectra (*Figure 6—figure supplement 3*) and unfolded with a single cooperative transition (*Figure 6—figure supplements 4–6*), allowing us to determine melting temperatures ($T_m$) for irreversible thermal denaturation. The three individually deleterious mutations were all destabilizing, with changes in melting temperatures ($\Delta T_m$) relative to the parent Aichi/1968 NP that ranged from −3.6°C to −4.9°C. N334H was stabilizing ($\Delta T_m$ of 4.4°C), and adding N334H to each of the individual destabilized mutants restored their stability to roughly wild-type values (*Figure 6E*). M136I is also modestly stabilizing, and partially rescues all three individually deleterious mutations (*Figure 6—figure supplements 4–7*). The final Brisbane/2007 NP contains two of the three identified destabilizing mutations (L259S reverts) and both of the identified stabilizing mutations—combining all four mutations in the parental Aichi/1968 background gives wild-type levels of activity (*Figure 6—figure supplement 8*).

These results suggest that most of the epistasis that we identified during NP's evolution is due to counterbalancing stabilizing and destabilizing mutations. To obtain a more complete portrait, we measured the stabilities of all of the resolved evolutionary intermediates (*Figure 6—figure supplements 4–6*). *Figure 7A* shows the correlation between activity and stability for all NP variants for which both properties were measured. For variants with melting temperatures exceeding 43°C, activity is independent of stability—changes in stability above this threshold are neutral with respect to activity. But once stability begins to fall below 43°C, there is a rapid decline in activity. A similar pattern is observed in the correlation between stability and viral growth, with growth declining precipitously once stability drops below 43°C (*Figure 7B*). The exception to this relationship is that variants containing R384G without E375G exhibit reduced growth even when they possess adequate stability and activity (*Figures 4B, 7A,B*). E375G is modestly destabilizing ($\Delta T_m = −1.0$°C), and so epistatically interacts with R384G by a mechanism other than protein stability. E375G and R384G induce opposite charge changes and occur on the same surface of NP (*Figure 5* and *Figure 7—supplement figure 1*)—we hypothesize that maintenance of the electrostatic charge on this surface might be important for NP's interaction with some partner late in the viral life cycle after RNA transcription is complete.

This caveat about R384G and E375G notwithstanding, it is striking that we can explain all of the other observed mutational effects simply in terms of protein stability. *Figure 7C* shows the trajectory of stability during NP's evolution. The parent Aichi/1968 is only marginally more stable than the

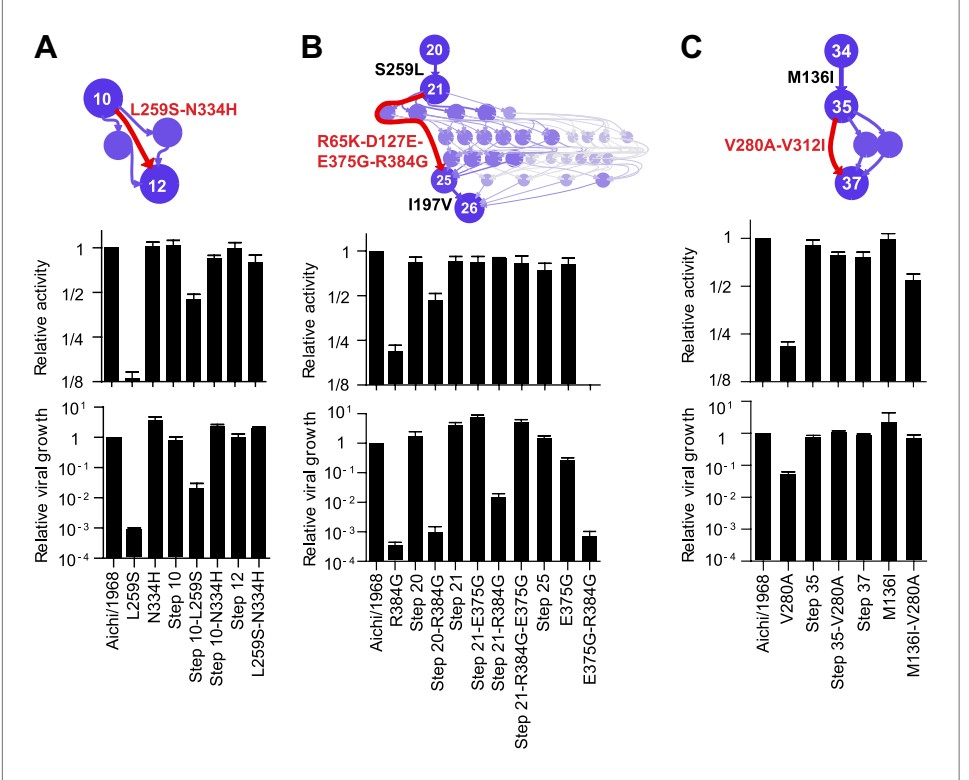

**Figure 4**. Effects of the individually deleterious mutations in the evolutionary intermediates in which they occurred. (**A**) L259S impairs the transcriptional activity and viral growth of both the parent Aichi/1968 and the evolutionary intermediate Step 10, but is rescued by N334H in both backgrounds. N334H alone has little effect on activity or growth in either background. The actual evolutionary trajectory involved the fixation of L259S and N334H in an unknown order. (**B**) R384G impairs activity and ablates growth of Aichi/1968, but has no effect on activity and a reduced adverse effect on growth in the high-confidence evolutionary intermediate (Step 21) in which it and several other mutations occurred in an unknown order. Addition of E375G to Step 21 with R384G fully rescues viral growth, but E375G alone worsens the impact of R384G. The reversion of L259S that preceded Step 21 plays an important role in enabling R384G, as the evolutionary intermediate without this reversion (Step 20) is more negatively affected by R384G. (**C**) V280A is deleterious in Aichi/1968 but not in the Step 35 evolutionary intermediate in which it actually occurred. M136I, which precedes V280A, largely rescues its effect.

minimal threshold where activity and viral growth begin to suffer. For this reason, the three destabilizing mutations L259S, R384G, and V280A are highly deleterious to Aichi/1968. During NP's evolution, stability fluctuates as the protein fixes stabilizing and destabilizing mutations. Each of the three destabilizing mutations that we identified as being under epistatic constraint is closely associated with a stabilizing mutation. The stabilizing M136I preceded the destabilizing V280A, and provided a stability cushion to eliminate V280A's otherwise deleterious effect (**Figures 4C, 7C**). The stabilizing S259L preceded the destabilizing R384G, and was necessary (in conjunction with E375G) to alleviate R384G's otherwise deleterious effect (**Figures 4B, 7C**). The stabilizing N334H occurred in close temporal proximity to the destabilizing L259S and fully counteracts L259S's otherwise deleterious effect (**Figures 4B, 7C**)—although in this case there is insufficient natural sequence data to determine which of these mutations occurred first (it is also possible that they occurred simultaneously).

## The epistatically constrained mutations contribute to viral immune escape

The aforementioned results illuminate the evolutionary steps that gave rise to the fixation of the individually deleterious destabilizing mutations, but they do not provide any indication of what forces may have

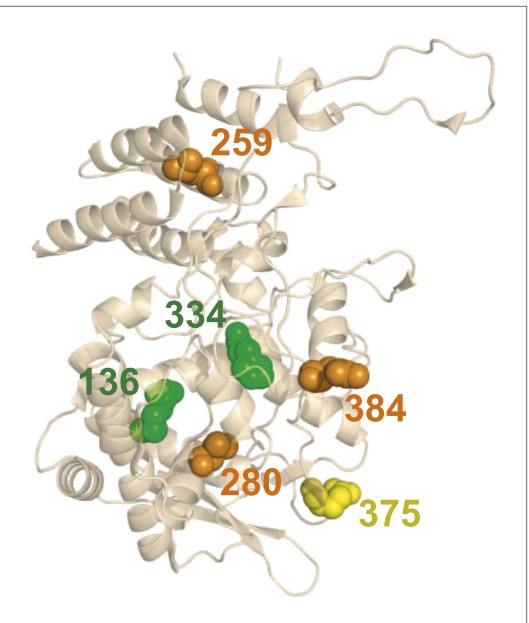

**Figure 5**. There is no obvious structural basis for the observed epistasis, as none of the epistatically interacting mutations are in contact in the solved crystal structures of NP. Shown above is one monomer from PDB structure 2IQH; the mutations are also not in contact in any of the known oligomeric structures (*Figure 5—figure supplement 1*). The sites of the three individually deleterious mutations are in orange, those of the rescuing mutations are in green, and the site of E375G (which can rescue R384G depending on genetic background) is in yellow.

The following figure supplements are available for figure 5:

**Figure supplement 1**. None of the epistatically interacting mutations are in contact in the known oligomeric structures of NP.

driven this fixation. The destabilizing mutations could have been fixed stochastically by genetic drift or hitchhiking, or they could have been directly favored by selection for viral immune escape. As discussed in the Introduction, NP is a target of CTLs, and mutations in CTL epitopes benefit influenza by helping it evade immune memory that accumulates in the human population (*Berkhoff et al., 2007*; *Valkenburg et al., 2011*).

We began by searching the literature for experimentally validated human CTL epitopes in NP. All three destabilizing mutations occur in characterized epitopes (*DiBrino et al., 1995*; *Voeten et al., 2000*; *Rimmelzwaan et al., 2004b*; *Berkhoff et al., 2007*; *Assarsson et al., 2008*; *Alexander et al., 2010*; *Cheung et al., 2012*), and mutations at two of these sites have been shown to reduce CTL recognition (*Voeten et al., 2000*; *Berkhoff et al., 2004*; *Rimmelzwaan et al., 2004b*; *Berkhoff et al., 2007*; *Figure 8—source data 1*).

To test if the epistatically constrained mutations are in more epitopes than expected by chance, we considered two approaches to comprehensively identify epitopes in NP: mining of a database of literature-characterized epitopes (*Vita et al., 2010*), and computational prediction of epitopes from protein sequence (*Stranzl et al., 2010*). The first approach has the advantage of only identifying experimentally validated epitopes, but the disadvantage that this set of epitopes is subject to unknown biases due to experimental choices about HLA types and viral strains. Computational prediction has the advantage of being unbiased with respect to HLA types and viral strains, but the disadvantage that the predictions may not be accurate.

As it turns out, both approaches give the same result—the three epistatically constrained mutations are significantly enriched in CTL epitopes relative to all sites in NP and to the set of sites that experienced substitutions along the evolutionary trajectory (*Figure 8* and *Figure 8—figure supplement 2*). These three destabilizing mutations are thus disproportionately important for viral immune escape, and may have been favored by selection for this property (a dN/dS test [*Murrell et al., 2013*] is inconclusive, probably due to lack of sequence data; *Figure 8—figure supplement 3* and *Figure 8—source code 2*). Stability-mediated epistasis therefore constrains the adaptive process of CTL escape as well as the sequence evolution of NP. The destabilizing CTL-escape mutations L259S, R384G, and V280A were inaccessible to NP during much of its evolutionary trajectory, but were fixed after stabilizing mutations made the protein permissive to their occurrence.

## Discussion

Our results paint a remarkably coherent picture of epistasis in NP evolution. We identified three mutations that are strongly deleterious to the original parent, yet were eventually fixed without adverse effect. All three mutations decrease NP's thermal stability. This decreased stability reduces in vivo protein levels, in turn reducing total transcriptional activity and viral growth. On the other hand, stabilizing mutations have little effect on protein levels, activity, or growth in the background of the parent

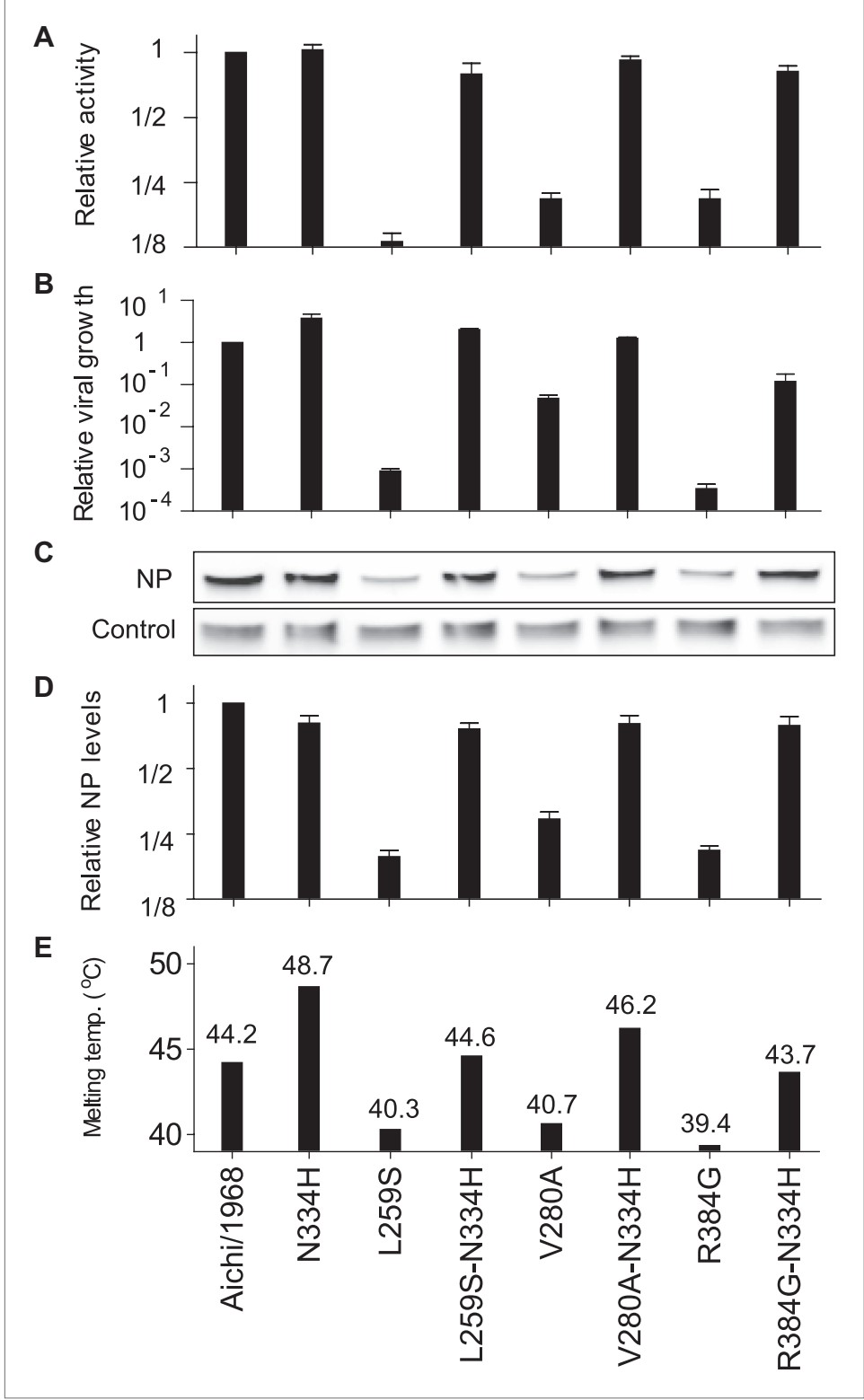

**Figure 6**. The epistasis correlates with mutational effects on NP stability. (**A**) and (**B**) N334H rescues activity and mostly rescues viral growth of all three individually deleterious mutations. (**C**) Western blot showing that the three individually deleterious mutations all reduce NP levels in transfected cells relative to an mCherry control expressed from an IRES in the same plasmid; this effect is counteracted by N334H. (**D**) Quantification of NP levels from triplicate Western blots (**Figure 6—figure supplement 1**). (**E**) The deleterious mutations decrease and N334H

*Figure 6. Continued on next page*

*Figure 6. Continued*

increases the stability of NP, as measured by thermal denaturation of purified protein monitored by circular dichroism (***Figure 6—figure supplements 2–6*** and ***Figure 6—source data 1***). ***Figure 6—figure supplement 7*** shows that M136I, which precedes V280A in the natural evolution, and is modestly stabilizing (***Figure 6—figure supplements 4–7***), also partially rescues the levels of V280A NP in transfected cells and the activity of all three individually deleterious mutations. Together, the two stabilizing mutations N334H and M136I can rescue the activity of combinations of the individually deleterious mutations (***Figure 6—figure supplement 8***).

The following source data and figure supplements are available for figure 6:

**Source data 1**. A table of all of the melting temperatures and the changes in stability relative to Aichi/1968, in CSV format.

**Figure supplement 1**. The full set of triplicate Western blots used to quantify the NP levels in transfected cells.

**Figure supplement 2**. Purification of Aichi/1968 NP with deletion of residues 2–7, mutation R416A, and a C-terminal 6-His tag (expected molecular weight 56.6 kDa).

**Figure supplement 3**. Circular dichroism wavelength scans for all variants that were tested (those with thermal melts shown in ***Figure 6—figure supplements 4–6***).

**Figure supplement 4**. The first 15 thermal denaturation curves.

**Figure supplement 5**. The second 15 thermal denaturation curves.

**Figure supplement 6**. The last 13 thermal denaturation curves.

**Figure supplement 7**. M136I partially rescues the activity of the three individually deleterious mutations, and mostly rescues protein levels for V280A.

**Figure supplement 8**. Activities for NPs with combinations of the stabilizing and destabilizing mutations.

NP—presumably because this parent is already sufficiently stable for its cellular environment. However, these stabilizing mutations play a crucial evolutionary role by counteracting the destabilizing mutations and enabling them to fix during evolution.

Of course, we do not wish to caricature protein evolution by suggesting that all epistasis is mediated by stability. In principle, mutations can affect a multitude of properties of NP, including its homo-oligomerization, association with RNA and other proteins, and cellular transport. We do not suggest that these properties are unimportant. In fact, we observe an epistatic interaction between R384G and E375G that likely relates to the electrostatic charge on one of NP's surfaces. Our assays are also of finite sensitivity, and so may miss small effects that are still significant to natural selection. There is also the potential for epistasis between NP and other viral proteins, although we see no evidence for such epistasis here, since all NP evolutionary intermediates that we tested are functional in a fixed background of other proteins. But doubtless some of these other mechanisms of epistasis would become apparent if we examined even longer evolutionary trajectories. However, the overriding message from our experiments is that stability-mediated epistasis is the dominant constraint on NP evolution.

Epistatically interacting mutations can be fixed in several ways. The mutations can accumulate sequentially without ever passing through a low-fitness intermediate (as in Maynard Smith's analogy), an initial deleterious mutation can be compensated by a subsequent mutation, or multiple mutations can occur simultaneously. We have identified two instances that clearly conform to Maynard Smith's paradigm: the stabilizing M136I preceded V280A, and the stabilizing S259L preceded R384G (***Figure 4B,C***). We also identified two instances where the actual evolutionary path is unclear due to a lack of natural sequence data from the relevant timeframe: N334H/L259S, and E375G/R384G (***Figure 4A,B***). However, in both cases it is at least possible that evolution conformed to Maynard Smith's paradigm: no simultaneous mutations or deleterious intermediates need have occurred if N334H preceded L259S, and if E375G preceded R384G (***Figure 4A,B***).

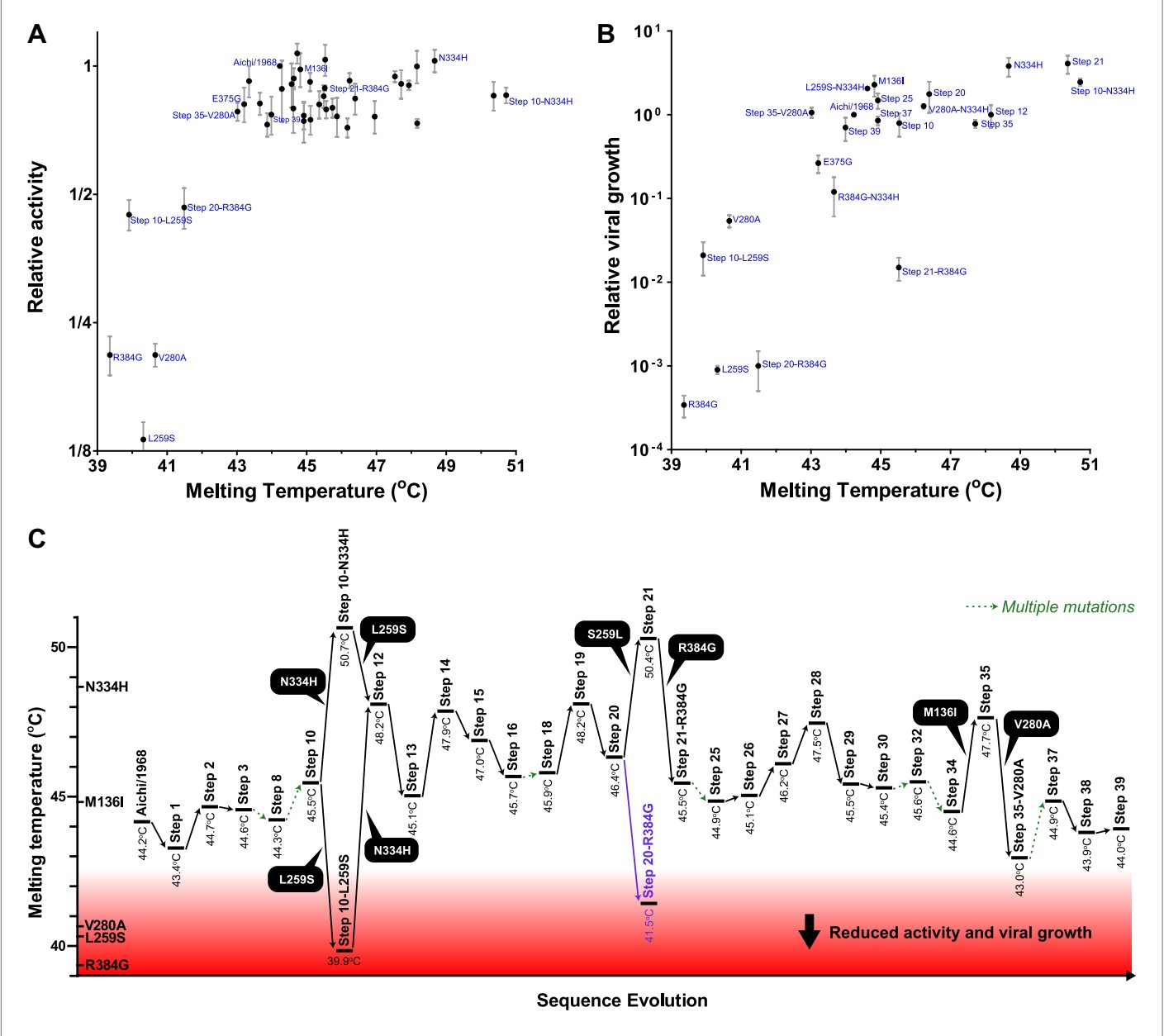

**Figure 7**. Most of the epistasis that we identified in NP's evolution can be explained by counterbalancing stabilizing and destabilizing mutations. (**A**) The relationship between NP stability and transcriptional activity for all variants for which both properties were measured. As long as the stability is greater than a threshold around 42°C, changes in stability are neutral with respect to activity. Below this threshold, activity declines sharply with decreasing stability. (**B**) The relationship between viral growth and NP stability exhibits a similar behavior. The exception is that growth of variants with R384G is fully rescued only by combining a stabilizing mutation with E375G (**Figure 4B**), an effect that we hypothesize is related to the electrostatic charge on one of NP's surfaces (**Figure 7—figure supplement 1**). (**C**) The dynamics of protein stability during NP evolution. Shown are the measured stabilities for evolutionary intermediates from the trajectory in **Figure 2A**. The lines along the y-axis at the far left show the stabilities of the five indicated individual point-mutants of the Aichi/1968 NP. Although the destabilizing mutations L259S, R384G, and V280A are deleterious to the Aichi/1968 parent, during evolution they are counterbalanced by stabilizing mutations. In the top panels, selected points are labeled with the NP variant name; the full data plotted in this figure are in **Figure 7—source data 1**.

The following source data and figure supplements are available for figure 7:

**Source data 1**. The activity, viral growth, and stability data shown in **Figure 7**.

**Figure supplement 1**. We hypothesize that E375G helps counteract R348G by maintaining the electrostatic charge on one of NP's surfaces.

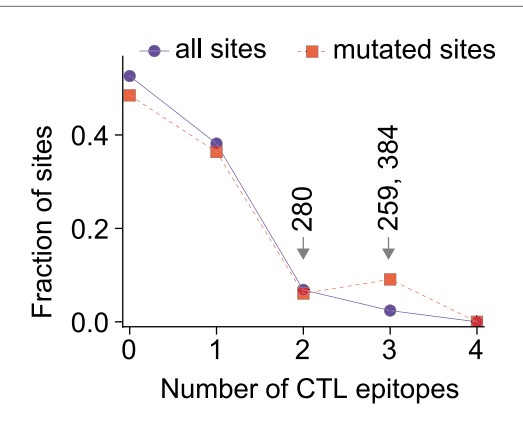

**Figure 8**. The three epistatically constraining destabilizing mutations occur at sites significantly enriched in human CTL epitopes. Distributions of the numbers of experimentally characterized epitopes per residue for all sites or sites that experienced mutations along the evolutionary trajectory are in blue and red, respectively. Sites 259, 280, and 384 are in significantly more epitopes than three random positions from all sites (p=0.001) or the mutated sites (p=0.004); however, three random positions from the mutated sites are not in significantly more epitopes than three random positions from all sites (p=0.157). Epitopes with experimentally characterized CTL responses were mined from the Immune Epitope Database (**Figure 8—source code 1** and **Figure 8—figure supplement 1**). The primary citation and summary information for epitopes involving sites 259, 280, and 384 are in **Figure 8— source data 1**. Similar results are obtained if CTL epitopes are instead predicted computationally (**Figure 8—figure supplements 1, 2**). A dN/dS analysis is inconclusive about whether the three sites are under positive or negative selection, probably due to lack of sequence data (**Figure 8—figure supplement 3** and **Figure 8—source code 2**).

The following source data, source code and figure supplements are available for figure 8:

**Source data 1**. Literature-characterized NP human CTL epitopes that include residues 259, 280, or 384 and contain sequences conserved or nearly conserved in Aichi/1968.

**Source code 1**. The input data files and the custom Python scripts used to identify human CTL epitopes in NP.

**Source code 2**. The data and source code used for the dN/dS analysis.

**Figure supplement 1**. Distribution of human CTL epitopes along the NP primary sequence.

*Figure 8. Continued on next page*

Previous experimental studies have identified stabilizing mutations as contributing to the evolution of enzyme specificities (**Bloom et al., 2006**), bacterial (**Wang et al., 2002**; **Bershtein et al., 2006**; **Weinreich et al., 2006**) and viral (**Chang and Torbett, 2011**) drug resistance, and H5N1 transmissibility (**Imai et al., 2012**). At a broader level, analyses of large datasets have shown that it is common for mutations to be deleterious to one protein homolog but benign to another (**Kondrashov et al., 2002**; **Baresic et al., 2010**). Our work illustrates how the dynamics of stability during evolution might explain these findings. As shown in **Figure 7C**, most of the intermediates during NP's evolution are only marginally more stable than the minimal threshold where function begins to suffer. This marginal stability of natural proteins has been noted previously, and been given two distinct explanations. The first explanation holds that evolution actively selects for marginal stability because both insufficient and excess stability are deleterious (**DePristo et al., 2005**; **Tokuriki and Tawfik, 2009**). The second explanation holds that evolution only selects against insufficient stability, but that proteins typically are marginally stable because most mutations are destabilizing and so extra stability is rapidly eroded by functionally neutral but destabilizing mutations (**Taverna and Goldstein, 2002**; **Bloom et al., 2007**). Our results decisively favor the second explanation for NP, since the evolving protein is usually marginally stable despite the fact that higher stability has no deleterious effect (**Figure 7A,B**). We therefore suggest the following: functionally neutral but stabilizing mutations occasionally fix by stochastic forces such as genetic drift, population bottlenecks, or hitchhiking. These stabilizing mutations enable NP to tolerate otherwise deleterious destabilizing mutations. Although these destabilizing mutations could in principle also fix by stochastic forces, the three that we have identified are actually adaptive since they contribute to viral immune escape. Stability-mediated epistasis therefore constrains NP's adaptation as well as its sequence evolution, since the accessibility of immune-escape mutations is dependent on the acquisition of enabling mutations.

It is intriguing to speculate whether similar forms of epistasis might constrain the evolution of other proteins. For example, the antigenic evolution of influenza hemagglutinin is punctuated, with a fairly constant rate of sequence change nonetheless leading to periodic jumps in antigenicity that require reformulation of the annual influenza vaccine (**Smith et al., 2004**). One

*Figure 8. Continued*

**Figure supplement 2**. The three epistatically constrained mutations are at sites significantly enriched in CTL epitopes as predicted by a computational epitope prediction program NetCTLpan1.1.

**Figure supplement 3**. It is largely inconclusive whether the sites of the three epistatically constrained mutations are under positive or negative selection as quantified by dN/dS values.

explanation that has been posited for this punctuated pattern is that adaptive antigenic change is limited not by the overall rate of substitution, but rather by the waiting time for the protein to accumulate antigenically neutral mutations that can be productively combined with mutations causing large antigenic changes (*Koelle et al., 2006*; *van Nimwegen, 2006*). Stability-mediated epistasis of the type that we have observed for NP provides at least one plausible mechanistic explanation for this and other cases of constrained molecular evolution.

## Materials and methods

### Evolutionary trajectory

Via Markov chain Monte Carlo implemented in BEAST (*Drummond et al., 2012*), we estimated the joint posterior distribution of phylogenetic trees and mutations along the branches of these trees given 431 date-stamped human H3N2 NP protein sequences from the Influenza Virus Resource. We assumed a Jones—Taylor—Thornton model (*Jones et al., 1992*) of protein substitution, a strict molecular clock and a relatively uninformative coalescent-based prior on the tree. We inferred the unobserved mutations via a data augmentation procedure that exploits uniformization and is robust to model misspecification (*Minin and Suchard, 2008*; *O'Brien et al., 2009*). *Figure 2B* reports the maximum clade credible tree from the posterior distribution.

Novel to this work, for each posterior sample, we converted the order of mutations along the line of descent from Aichi/1968 and Brisbane/2007 into a directed graph through sequence space (*Figure 2—figure supplement 1*). Summarizing these graph samples effectively integrates over uncertainty in the tree and substitution process, returning the marginal posterior distribution of the evolutionary trajectory of interest. In our GraphViz visualization, each circle represents a unique inferred sequence. Areas and intensities of circles are proportional to the posterior probability that the true trajectory visited that sequence. Lines correspond to mutations, with thickness and intensity proportional to the posterior probability that specific mutation connected those two sequences. We labeled connections with posterior probability ≥60% in black in *Figure 2*; mutations lacking high-confidence connections are in red. Likewise, we considered sequences with posterior probability ≥60% as high confidence and assigned them numeric labels. *Figure 2—source code 1* contains the relevant computer code.

### Transcriptional activity

We reverse-transcribed the Aichi/1968 NP and the Nanchang/1995 PB2, PB1, and PA from viral RNA (BEI Resources NR-9534 and NR-3222), and cloned them into pHW2000 (*Hoffmann et al., 2000*) (a gift from Y Kawaoka) to create pHWAichi68-NP, pHWNan95-PB2, pHWNan95-PB1, and pHWNan95-PA (*Supplementary file 1*). Similar plasmids were constructed for the PB2, PB1, and PA of Aichi/1968 and Brisbane/2007 and named pHWAichi68-PB2, pHWAichi68-PB1, pHWAichi68-PA, pHWBR07-PB2, pHWBR07-PB1, and pHWBR07-PA (*Supplementary file 1*). We used site-directed mutagenesis to create the other NP variants (*Supplementary file 2*). Note that the final sequence at the end of our trajectory (Step 39) matches the Brisbane/2007 protein sequence, but does not match the nucleotide sequence of this strain as we did not introduce any of the synonymous mutations.

We measured activity using a previously described (*Bloom et al., 2010*) reporter plasmid encoding a GFP vRNA with termini from the A/WSN/1933 PB1. We co-transfected this reporter into 12-well dishes of 293T cells along with 50 ng of NP plasmid and 200 ng each of pHWNan95-PB2, pHWNan95-PB1, and pHWNan95-PA (*Figure 3—figure supplement 1*). This amount of NP plasmid is near the midpoint of the dose-response curve (*Figure 3—figure supplement 1*). After 20 hr, we quantified the GFP mean-fluorescence intensity (MFI) by flow cytometry. We seeded 293T cells at $2 \times 10^5$ per well 20–24 hr pre-transfection in D10 (DMEM with 10% heat-inactivated fetal bovine serum, 2 mM L-glutamine, 100 U/ml penicillin, and 100 µg/ml streptomycin). We quantified the activity relative to the average

for three replicates of wild-type pHWAichi68-NP. We performed at least three biological replicates for each variant, with each replicate performed on a different day using an independent plasmid mini-prep. *Figure 3—source data 1* gives the means and standard errors of the activities for all NP variants.

## Viral growth

We grew viruses carrying GFP in the PB1 gene using a modification of a previously described system (*Bloom et al., 2010*; *Figure 3—figure supplement 2*). We used lentiviral transduction to create the cell lines 293T-CMV-Nan95-PB1 and MDCK-SIAT1-CMV-Nan95-PB1, which express the coding sequence of the Nanchang/1995 PB1 with the F2 peptide disrupted after its eighth codon (*Chen et al., 2001*) under control of a CMV promoter. We seeded co-cultures of $2 \times 10^5$ 293T-CMV-Nan95-PB1 and $2 \times 10^4$ MDCK-SIAT1-CMV-Nan95-PB1 cells in D10 media in six-well dishes, and 20–24 hr later transfected with 250 ng each of pHWNan95-PB2, pHH-PB1flank-eGFP (*Bloom et al., 2010*), pHWNan95-PA, a pHWAichi68-NP variant, pHW184-HA, pHW186-NA, pHW187-M, and pHW188-NS. These last four plasmids (*Hoffmann et al., 2000*) encode genes from A/WSN/1933 (gifts from Y Kawaoka). After 20–24 hr, we replaced the D10 with influenza growth media (Opti-MEM I with 0.3% BSA, 0.01% heat-inactivated fetal bovine serum, 100 U/ml penicillin, 100 µg/ml streptomycin, and 100 µg/ml calcium chloride). After 66 hr (shortly before peak titers, *Figure 3—figure supplement 2*), we collected the supernatant and clarified it for 5 min at 2000×$g$. We infected dilutions of supernatant into 12-well dishes seeded 8 hr earlier at $10^5$ MDCK-SIAT1-CMV-Nan95-PB1/well in influenza growth media, and 16 hr later determined the titer by flow cytometry by using the Poisson equation to estimate the viral titer based on the fraction of GFP positive cells. We performed these titering infections with various volumes of viral supernatant such that the titer could be computed from an infection with between 0.5% and 10% of cells green. Note that these titers reflect the number of particles that are able to productively infect cells and transcribe high levels of GFP from viral RNA—they may not reflect the same titers as would be determined using other approaches such as plaque assays or tissue-culture infectious dose 50% assays. We quantified the titer relative to the average of three replicates of the wild-type pHWAichi68-NP (typically around $10^3$ infectious particles per microliter). We performed at least three biological replicates for each variant, with each replicate performed on a different day using an independent plasmid mini-prep. *Figure 3—source data 2* gives the means and standard errors.

## NP protein levels

We cloned the NP coding sequence into a mammalian expression plasmid under control of a CMV promoter with a FLAG tag inserted between the N-terminal methionine and the second residue. After the NP stop codon, we added an internal ribosome entry site (IRES) followed by mCherry with a C-terminal FLAG tag. We seeded 293T cells at $2 \times 10^5$ per well in D10 in 12-well dishes, and 20–24 hr later transfected with 400 ng of plasmid. After 20 hr, we collected the cells and lysed them on ice in 100 µl of RIPA buffer with one protease-inhibitor tablet (Roche, Basel, Switzerland, 05892791001) per 10 ml. We pelleted debris at 21,000×$g$ for 10 min, and loaded 2.5 µl of clarified supernatant on an SDS-PAGE gel after boiling with a reducing sample-loading buffer. We transferred the protein to a PVDF membrane and stained with a 1:5000 dilution of mouse anti-FLAG (Sigma, St. Louis, MO, F1804) followed by a 1:2500 dilution of Alexa Flour 680-conjugated goat anti-mouse (Invitrogen, Grand Island, NY, A-21058), using Li-Cor Odyssey (Lincoln, NE) blocking buffer (Li-Cor 927-40,000) and performing washes with TBS-T (Pierce, Rockford, IL, 28360). We quantified the ratio of NP to the corresponding mCherry control using a Li-Cor Odyssey Infrared Imaging System, and normalized this ratio to that for wild-type Aichi/1968 NP (*Figure 6—figure supplements 1, 7*).

### Protein stability

In order to obtain non-aggregated RNA-free NP in a CD-compatible buffer, we introduced two previously described (*Ye et al., 2006*) modifications: deletion of residues 2–7 and R416A. We cloned NP with these modifications and a C-terminal 6-histidine tag into pET28b(+), transformed into BL21 Star DE3 (Invitrogen, Grand Island, NY, C6010-03), and grew 1 L of these bacteria to an OD600 of 0.3–0.6 at 37°C. We then chilled the cultures on ice, reduced the shaker temperature to 20°C and induced with 500 µM IPTG. After overnight growth, we pelleted the cells and lysed them on ice by sonication in 50 ml of 50 mM sodium phosphate pH 8.0, 500 mM sodium chloride, 0.5% Triton X-100, 10 mM imidazole, 1 mM PMSF, 0.1 mg/ml magnesium chloride, 1 mM lysozyme, and 1000 units of benzonase

(Sigma, St. Louis, MO, E1014). We clarified the supernatant for 30 min at 10,000×g and 4°C, passed it through a 0.45-μm filter, and purified NP over a cobalt column (Pierce, Rockford, IL, 89969) using the manufacturer's protocol but eluting with 200 mM imidazole. We concentrated the protein with an Amicon Ultra 30 kDa filter, and dialyzed it against CD buffer (20 mM sodium phosphate pH 7.0 with 300 mM sodium fluoride) in a 20-kDa dialysis device. We further purified the protein over a Superdex 200 GL size-exclusion column. All variants eluted in a single monomeric peak, and all had ratios of absorbance at 260 nm to absorbance at 280 nm of less than 0.65 (*Figure 6—figure supplement 2*).

We diluted the proteins to 5 μM in CD buffer as determined by the absorbance at 280 nm in a 1 cm quartz cuvette using an extinction coefficient of 0.0566 μM/cm, and acquired CD spectra at 20°C with a Jasco J-815 spectropolarimeter. All variants exhibited similar spectra with typical alpha-helical characteristics (*Figure 6—figure supplement 2, 3*). We performed thermal melts at a scan rate of 2°C per minute, monitoring ellipticity at 209 nm. All variants unfolded with a single cooperative transition, allowing us to obtain melting temperatures from sigmoidal curve fits (*Figure 6—figure supplements 4–6* and *Figure 6—source data 1*). The melting was irreversible (*Figure 6—figure supplement 2*), preventing us from calculating equilibrium thermodynamic stabilities.

## CTL epitopes

We identified human CTL epitopes of ≤12 residues with at least 89% conservation in Aichi/1968 NP and a verified T-cell response from the Immune Epitope Database (*Vita et al., 2010*). *Figure 8—source data 1* lists primary citations for epitopes involving residues 259, 280, and 384. *Figure 8—figure supplement 1* shows the number of epitopes at each position. We computed p-values by randomly drawing three different residues from the set of all sites or all mutated sites, and comparing the number of epitopes for these sites to the number for sites 259, 280, and 384 (*Figure 8*). p-values represent the fraction of $10^5$ random draws that contained at least as many epitopes as sites 259, 280, and 384. We performed a similar analysis for epitopes predicted by NetCTLpan 1.1 (*Stranzl et al., 2010*) using the default settings for 9-mer peptides and the HLA supertypes (*Figure 8—figure supplement 2*). The data and computer code are in *Figure 8—source code 1*.

The dN/dS comparisons shown in *Figure 8—figure supplement 3* were performed with FUBAR (*Murrell et al., 2013*) using the DataMonkey server. The sequence data, results, and analysis are in *Figure 8—source code 2*.

## Acknowledgements

We thank J Bolduc, B Stoddard, M Daugherty, S Schetterer, and S Ovchinnikov for assistance and advice with protein purification; K Blair and L Warfield for assistance with Western blotting; C Correnti and R Strong for assistance and advice with circular dichroism; T Hertz and A Gartland for advice about identifying CTL epitopes; Y Kawaoka for reverse-genetics plasmids; and the BEI Resources program for viral RNA used to clone genes.

## Additional information

### Funding

| Funder | Grant reference number | Author |
| --- | --- | --- |
| National Institutes of Health | 1 R01 GM102198-01 | Lizhi Ian Gong, Jesse D Bloom |
| National Institutes of Health | 1 R01 GM086887-05 | Marc A Suchard |

The funders had no role in study design, data collection and interpretation, or the decision to submit the work for publication.

### Author contributions

LIG, Performed the experiments, analyzed the data, and wrote the manuscript; MAS, Developed the probabilistic techniques and performed the computational work, analyzed the data, and wrote the manuscript; JDB, Designed the study, performed the computational work, analyzed the data, wrote the manuscript

## Additional files

### Supplementary files

• Supplementary file 1. The full vRNAs in the reverse-genetics plasmids used in this study. A text file giving the vRNAs inserted between the RNA polymerase I promoter and terminator in the reverse-genetics plasmids pHWAichi68-NP, pHWNan95-PB2, pHWNan95-PB1, pHWNan95-PA, pHWAichi68-PB2, pHWAichi68-PB1, pHWAichi68-PA, pHWBR07-PB2, pHWBR07-PB1, pHWBR07-PA, pHW184-HA, pHW186-NA, pHW187-M, pHW188-NS, and pHH-PB1flank-eGFP.

• Supplementary file 2. The protein-coding sequences of all NP variants used in this study. A text file giving the coding sequences for the NP variants used in this study.

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
