## [Decision Letter]

Thank you for sending your work entitled “Stability-mediated epistasis constrains the evolution of an influenza protein” for consideration at *eLife*. Your article has been favorably evaluated by a Senior editor and 3 reviewers, one of whom is a member of our Board of Reviewing Editors. The following individuals responsible for the peer review of your submission want to reveal their identity: Mercedes Pascual (Reviewing editor) and Fyodor Kondrashov (peer reviewer). The Reviewing editor and the other reviewers discussed their reviews before we reached this decision, and the Reviewing editor has prepared the follow comments to help you prepare a revised submission.

This work combines computational and experimental methods to demonstrate stability-mediated epistasis in the evolutionary trajectory of the influenza virus. It provides the first experimental exploration of fitness landscapes in actual evolutionary trajectories in sequence space.

The authors experimentally dissect the substitutions that have accrued in a naturally evolving influenza protein to argue that most of epistasis among sites is mediated by stability, and that stabilizing mutations are often pre-requisites for adaptive substitutions, which would otherwise be too destabilizing for a protein to accommodate. Although the referees felt that this was an ambitious and remarkable study in many ways, they also had a number of comments, summarized below, meant to strengthen the evidence and provide additional connections to existing literature and theory.

1) The Introduction and Discussion are written both from the perspective of protein evolution in general terms and evolution of NP in particular. For readers not already familiar with the background on the subject, it would be useful to better place the study in the context of previous work, so that its novelty can be more clearly conveyed. In particular, the authors rely on the concept of substitutions occurring one after another, as envisioned by Maynard Smith. This is well suited for exploring the concept of epistasis but does not capture all possibilities. Specifically, an alternative to sequential fixation of substitutions, compensatory substitutions (including instances when two individually deleterious substitutions are neutral when combined), being fixed at the same time, has been proposed by Kimura (1985, J. Genet 64: 7-19), with some evidence of this particular route of compensatory evolution found in tRNAs (Meer, 2010 Nature 464:279-82). In the authors’ data, two out of the three individual substitutions appear to have been fixed by Kimura’s model rather than Maynard Smith’s. Kimura’s model should be mentioned in the Introduction and also in the part that interprets the evolutionary mechanism at work in the fixation of L259S and R384G. To our knowledge, epistasis of the Kimura type has never been described for proteins, which would make this study even more exciting.

Also, the issue of structural stability as being an important factor in compensatory interactions has been discussed before (see Kondrashov AS, Sunyaev S, Kondrashov FA. Dobzhansky-Muller incompatibilities in protein evolution. Proc Natl Acad Sci U S A. 2002 Nov 12;99(23):14878-83; DePristo MA, Weinreich DM, Hartl DL. Missense meanderings in sequence space: a biophysical view of protein evolution. Nat Rev Genet. 2005 Sep;6(9):678-87; and Tóth-Petróczy A, Tawfik DS. Slow protein evolutionary rates are dictated by surface-core association. Proc Natl Acad Sci U S A. 2011 Jul 5;108(27):11151-6).

2) The study is described as dissecting, within the noise limits of the assays, the full extent of epistasis along the evolutionary trajectory of NP. However, the authors assay each of 39 individual mutational effects in the context of the *ancestor* strain alone, as well as the specific mutations that help to alleviate three apparently deleterious mutations. Thus, there may be much more epistasis that has been important to the specific evolutionary trajectory that NP has taken than the authors have been able to assay, because the authors reconstruct mutations in the ancestral strain alone. Although we recognize the extent of the experimental work, there seems to be no good reason to distinguish the ancestral strain and look only for epistasis relative to mutational effects on that background. Consideration of a small (5?) additional number of reconstructed mutations in different backgrounds would strengthen the evidence for the authors’ conclusions, if none of these were to prove as destabilizing as the three that are at the center of the argument. We recognize that this additional work may take a considerable time and recommend that the authors consider this extension only if it can be pursued within a relatively short time. Alternatively, we recommend that the authors acknowledge the sources of epistasis that they are not testing for given their experimental design and temper their conclusions accordingly.

3) No direct evidence is presented for positive selection of the three mutations deleterious in the ancestral strain. The authors should consider estimating dN/dS values at these specific sites, to the extent that they have enough data to do so.

4) An alternative hypothesis to explain the experimental observations is that throughout the course of the evolutionary trajectory there is a pattern of destabilizing mutations followed by stabilizing mutations, simply so that NP can maintain a roughly consistent folding energy. If so, the patterns shown in Figure 6, of the stabilizing mutations that precede each of the three focal deleterious mutations, would not be unique to these three special epistatic mutations, but rather would occur for many pairs of subsequent mutations along the evolutionary trajectory. Additional melting experiments would therefore provide stronger evidence for the conclusion that epistasis permitting *adaptive* mutations is the primary reason for observing stabilizing/destabilizing pairs.

---

## [Author Response]

During further data analysis and experiments that we performed subsequent to our initial submission, we discovered technical problems with the assay that we used to quantify protein concentrations for our earliest stability measurements.

We have repeated the affected measurements at the proper protein concentration. In our revised manuscript, we have replaced all data from the affected measurements with our new measurements.

We stress that the changes in the measured stabilities are small and do not alter any of our conclusions, but they do slightly alter the numerical values of the stabilities. At the bottom of this letter, after the point-by-point response, we provide a detailed description of the original technical problem and the corrected measurements.

*1) The Introduction and Discussion are written both from the perspective of protein evolution in general terms and evolution of NP in particular. For readers not already familiar with the background on the subject, it would be useful to better place the study in the context of previous work, so that its novelty can be more clearly conveyed. In particular, the authors rely on the concept of substitutions occurring one after another, as envisioned by Maynard Smith. This is well suited for exploring the concept of epistasis but does not capture all possibilities. Specifically, an alternative to sequential fixation of substitutions, compensatory substitutions (including instances when two individually deleterious substitutions are neutral when combined), being fixed at the same time, has been proposed by Kimura (1985, J. Genet 64: 7-19), with some evidence of this particular route of compensatory evolution found in tRNAs (Meer, 2010 Nature 464:279-82). In the authors’ data, two out of the three individual substitutions appear to have been fixed by Kimura’s model rather than Maynard Smith’s. Kimura’s model should be mentioned in the Introduction and also in the part that interprets the evolutionary mechanism at work in the fixation of L259S and R384G. To our knowledge, epistasis of the Kimura type has never been described for proteins, which would make this study even more exciting*.

This comment is correct in that we over-emphasized the Maynard Smith paradigm (one mutation enabling a second without ever passing through a fitness valley). In several cases, our data are also consistent with two other paradigms: simultaneous mutations, or a deleterious mutation followed by a compensatory change. To address this issue, we have cited the Kimura and Meer papers. We have also revised relevant text, beginning with the Abstract where we replaced the statement “were preceded by stabilizing mutations” with “were preceded or accompanied by stabilizing mutations.” Finally, we have added a paragraph in the Discussion that discusses this issue.

However, we do not agree that our data preferentially support the Kimura’s paradigm over Maynard Smith’s in any of the cases. Figure 2 shows the evolutionary trajectory as it can be reconstructed from available influenza sequences. The regions in red (such as those involving L259S/N334H and R384G/E375G) do not necessarily imply that the mutations occurred simultaneously – they simply indicate that they occurred sufficiently close in time that no viral isolates were sequenced in the intervening time period, making it impossible to determine the actual order.

In the revised manuscript, we specify that for two of the four epistatically interacting mutations, the evidence definitively support Maynard Smith’s paradigm (M136I preceded V280A, and S259L preceded R384G) – but that in the other two cases, we cannot resolve the relative order. However, we note that in both ambiguous cases, our data (Figure 4) show that it is possible to construct an evolutionary path consistent with Maynard Smith’s paradigm in that the mutations could have fixed sequentially without ever causing a measurable decrease in NP function (if N334H preceded L259S in the background of Step 10, and if E375G preceded R384G in the background of Step 21).

*Also, the issue of structural stability as being an important factor in compensatory interactions has been discussed before (see Kondrashov AS, Sunyaev S, Kondrashov FA. Dobzhansky-Muller incompatibilities in protein evolution. Proc Natl Acad Sci U S A. 2002 Nov 12;99(23):14878-83; DePristo MA, Weinreich DM, Hartl DL. Missense meanderings in sequence space: a biophysical view of protein evolution. Nat Rev Genet. 2005 Sep;6(9):678-87; and Tóth-Petróczy A, Tawfik DS. Slow protein evolutionary rates are dictated by surface-core association. Proc Natl Acad Sci U S A. 2011 Jul 5;108(27):11151-6)*.

We thank the reviewers for pointing out these references. We find the comparison with the Kondrashov et al paper to be particularly interesting, since our study is really an investigation of “compensated pathogenic deviations” in Brisbane/2007 NP relative to Aichi/1968 NP. The frequency of individually deleterious mutations that we identify (3 out of 34 unique mutations) is similar to the 10% frequency of compensated pathogenic deviations reported by Kondrashov et al. For reasons of brevity and scope, we have not expanded on this connection in the current manuscript, but it would be interesting to follow up on this connection in the future. For now, we cite the Kondrashov et al paper in the Discussion to briefly touch on the issue.

The DePristo et al paper is now cited in the Discussion. However, we note that there is a clear difference between the view introduced in that paper and what we find for NP. DePristo et al argue that there is selection to keep stability within a narrow margin, with either insufficient or extra stability being deleterious. In contrast, we find no evidence that extra stability is deleterious to NP – as shown in Figure 7, high stability has no measurable negative effect. We therefore think that our results support only half of the connection posited in the DePristo et al paper.

We have not cited the Toth-Petroczy and Tawfik paper mentioned above, but we have cited another paper by the same senior author (Bershtein et al, Science, 2006) that we believe more directly relates to our manuscript.

*2) The study is described as dissecting, within the noise limits of the assays, the full extent of epistasis along the evolutionary trajectory of NP. However, the authors assay each of 39 individual mutational effects in the context of the ancestor strain alone, as well as the specific mutations that help to alleviate three apparently deleterious mutations. Thus, there may be much more epistasis that has been important to the specific evolutionary trajectory that NP has taken than the authors have been able to assay, because the authors reconstruct mutations in the ancestral strain alone. Although we recognize the extent of the experimental work, there seems to be no good reason to distinguish the ancestral strain and look only for epistasis relative to mutational effects on that background. Consideration of a small (5?) additional number of reconstructed mutations in different backgrounds would strengthen the evidence for the authors’ conclusions, if none of these were to prove as destabilizing as the three that are at the center of the argument. We recognize that this additional work may take a considerable time and recommend that the authors consider this extension only if it can be pursued within a relatively short time. Alternatively, we recommend that the authors acknowledge the sources of epistasis that they are not testing for given their experimental design and temper their conclusions accordingly*.

This is an important point. Our approach will not find all epistasis – it just finds mutations that are deleterious in the Aichi/1968 NP when the other genes are held constant. There are two ways our approach could “miss” epistasis, neither of which can feasibly be experimentally addressed in a systematic way (we appreciate that the reviewers recognize this, and are simply asking us to respond as best we can). Since we are uncertain which shortcoming is being referenced in the above comment, we discuss both here. We have addressed one with additional experimental data and the other by revising the text.

The first shortcoming is that we are looking at mutations to NP in a fixed background of other influenza genes. It is possible that some NP mutations could be rescued by mutations in other genes, which also evolved between 1968 and 2007.

Experimentally reconstructing all mutations in all viral genes is not feasible. However, several facts indicate that most of the NP epistasis in the timeframe we examined is intra-genic rather then inter-genic. First, all of the naturally occurring NP evolutionary intermediates are functional in a fixed genetic background of other genes. Second, we can explain the epistasis in terms of NP-intrinsic properties (such as stability) that do not involve other viral genes. To more firmly support this case, we have added new data in Figure 3—figure supplement 3 that shows the effects of the three individually deleterious mutations (L259S, R384G, V280A) in three different backgrounds of polymerase genes: Aichi/1968, Nanchang/1995, and Brisbane/2007. As shown in this new figure supplement, the mutations are deleterious in all three backgrounds, supporting the idea that these mutations are constrained by epistasis inherent to NP itself.

The second shortcoming involves the approach we used to look for intra-genic epistasis in NP. It is possible that there are mutations that are not deleterious to either Aichi/1968 NP or the evolutionary intermediate in which they occurred (and so do not appear impaired in either Figure 3 or Figure 3), but are still deleterious in some other NP genetic background. So the reviewers are correct in stating that we have not found all epistasis – we have just devised a systematic test for a specific subset of epistasis. We have revised the text to acknowledge this point. For example, in the Introduction we now say, “While this experimental strategy is not guaranteed to find every possible epistatic interaction along the evolutionary trajectory, it will systematically identify all mutations that have different effects in the original parent and the evolutionary intermediates in which they actually occurred.”

However, we do not feel it is feasible to address this point by further experimentation. First, introducing each mutation in every genetic background would require a massive number of additional experiments. Second, as we proceed along the evolutionary trajectory, the mutations already start to exist in the evolutionary intermediates, so it is not even possible to introduce them. For example, one might ask – do the mutations have the same effect in the final Brisbane/2007 NP as they do in the initial Aichi/1968 NP? However, the mutations are already present in the Brisbane/2007 NP, so it is not possible to introduce them into this background. One could imagine reverting mutations out of the Brisbane/2007 NP, but this is a conceptually different experiment, as it no longer involves introducing mutations in a forward evolutionary direction. However, if any other un-identified epistasis is mediated by stability, then Figure 7 with the new data we have added (described below in response to comment 4) suggests that we found the most individually destabilizing mutations along the trajectory.

The Aichi/1968 parent is one of the least stable naturally occurring sequences, so is sensitive to destabilizing mutations. If we had started with a more stable parent (such as Step 19), then we would have missed many destabilizing mutations since this more stable parent would have tolerated mutations that were deleterious to Aichi/1968 NP.

*3) No direct evidence is presented for positive selection of the three mutations deleterious in the ancestral strain. The authors should consider estimating dN/dS values at these specific sites, to the extent that they have enough data to do so*.

In response to this comment, we have removed any specific references to “positive selection” and now merely make the well-supported claim that the three destabilizing mutations are enriched in experimentally characterized CTL epitopes and so likely to contribute to viral immune escape.

In our view, dN/dS values represent a statistical test that attempts to detect positive selection under certain assumptions. However, we believe it is preferable to use independent experimental evidence to determine whether a site is of value for immune escape (as we have done) rather than to rely on dN/dS. Typically papers that report methods for detecting positive selection (such as dN/dS) develop a statistical method and then validate it by testing whether it can identify sites with independent evidence for positive selection. We describe extensive experimental evidence culled from many independent citations that the sites in question are located in human CTL epitopes.

In addition, dN/dS tests are based on the idea that positively selected sites should change more rapidly than other sites (hence the elevation of dN over dS). But as we have shown, sites that contribute to CTL escape may be under strong constraint, and so may only be able to change under certain circumstance (in this case when they co-occur with stabilizing mutations), even if a change might be beneficial. This sort of epistasis would depress the rate of substitution at a site, reducing the signal in a dN/dS test even if a site is strongly targeted by CTLs. Nonetheless, we have performed dN/dS tests and added the results in Figure 8—figure supplement 3 and [Supplementary-material SD9-data]. The signal is weak, probably because of a lack of data. Although we have many hundreds of NP sequences, they are closely related and are derived from a phylogenetic tree with a trunk-like structure (Figure 2), which reduces the power of the analysis since the sequences all closely track a single line of descent. The three destabilizing mutations occur at sites for which the posterior probability of positive selection is higher than for the typical site in NP, but for none do the posterior probabilities of positive selection rise to >95%. But conversely, none of the destabilizing mutations occur at sites for which the posterior probability of negative selection is >80%, indicating that these sites are also unlikely to be under strong negative selection. The dN/dS test therefore is inconclusive about whether the sites are under positive or negative selection.

*4) An alternative hypothesis to explain the experimental observations is that throughout the course of the evolutionary trajectory there is a pattern of destabilizing mutations followed by stabilizing mutations, simply so that NP can maintain a roughly consistent folding energy. If so, the patterns shown in Figure 6, of the stabilizing mutations that precede each of the three focal deleterious mutations, would not be unique to these three special epistatic mutations, but rather would occur for many pairs of subsequent mutations along the evolutionary trajectory. Additional melting experiments would therefore provide stronger evidence for the conclusion that epistasis permitting* adaptive *mutations is the primary reason for observing stabilizing/destabilizing pairs*.

Our initial manuscript was overly strong in its statement of the association between the stabilizing/destabilizing mutations and the adaptive mutations. We have revised the text and added experimental data as described below.

Our results show that stabilizing mutations make NP tolerant to destabilizing mutations, and that in many cases these destabilizing mutations contribute to immune escape. However, our results do not imply causality in terms of adaptive mutations being the reason for the stabilizing/destabilizing pairs. Our results do, however, argue against a hypothesis that has often been raised in the literature: too much stability is bad, so selection for constant stability actively favors destabilizing mutations after a stabilizing mutation. The evidence against this hypothesis is Figure 7, which shows no measurable deleterious impact of extra NP stability – destabilizing a highly stable NP is neutral with respect to function, not beneficial.

We have elaborated on this issue with new experimental data: we have now measured the stabilities of all of the resolved evolutionary intermediates and added them to Figure 7. Previously we had just shown the stabilities of the evolutionary intermediates that immediately preceded or followed L259S, R384G, and V280A.

These new data show that stability indeed fluctuates during NP’s evolution. Although the largest decreases in stability are associated with the three mutations on which we focus (L259S, R384G, and V280A), other slightly destabilizing mutations are also fixed during NP’s evolution. We subscribe to the following interpretation: during its evolution, NP occasionally acquires stabilizing mutations that are neutral with respect to function (at least within the sensitivity of our experiments). Once these stabilizing mutations have occurred, the protein is tolerant to destabilizing mutations. It also remains tolerant to additional stabilizing mutations, but most frequently the next mutation will be destabilizing (it is well known that most random mutations are destabilizing).

Some of these destabilizing mutations are totally neutral, while others are neutral with respect to function but promote immune escape. These destabilizing mutations can fix by stochastic forces (drift, hitchhiking) or selective forces (immune escape). However, a mutation favored by selective forces will fix more frequently than one just driven just by stochastic forces, so the virus often fixes immune-escape mutations once they are accessible, explaining why the destabilizing mutations are enriched in CTL epitopes. After the protein has exhausted the extra stability provided by the original stabilizing mutation, selection disfavors further destabilizing mutations until the protein has again acquired a stabilizing mutation.

The explanation in the previous paragraph is our interpretation, but is too speculative to include directly in the manuscript text. We have included some more conservative mention of this issue in the Discussion. We have also re-written relevant portions of the text to emphasize that all we can conclusively claim is that there is a pattern of stabilizing/destabilizing mutations, and that the three most destabilizing mutations make disproportionate contributions to immune escape. For example, the last sentence of the original abstract read “… with stabilizing mutations permitting subsequent destabilizing but adaptive mutations.” We have changed this to read “… with stabilizing mutations permitting otherwise inaccessible destabilizing mutations which are sometimes of adaptive value.”

*Corrections to stability measurements performed at incorrect protein concentration*:

During work subsequent to submission of our original manuscript, we discovered that some of our early stability measurements were performed at a protein concentration below 5 μM. We have repeated these measurements and replaced the affected data with our new measurements. We stress that although some of the numbers are slightly different, the changes do not alter any of the conclusions (the destabilizing mutations are still destabilizing and the stabilizing mutations are still stabilizing).

This problem arose because for the first few protein variants that we purified, we quantified concentration using a Bradford assay that appears to have been unreliable. After these first few variants, we switched to quantifying the concentration using the absorbance at 280 nm.

Originally, we had not realized that the Bradford assay concentrations were inaccurate because we analyzed the melting curves in normalized units. However, subsequent to our initial submission, we examined an overlay of all the CD traces and discovered that some traces had smaller magnitudes. We looked back at our notes and realized these were the variants quantified with the Bradford assay. We therefore re-purified these variants and measured their stabilities at the correct concentration as determined by absorbance at 280 nm. For these new measurements, the magnitudes of the CD traces are all similar, indicating the concentrations were accurate. In most cases, the stability measurements at the correct concentrations were higher by a fraction of a degree, suggesting slight concentration dependence to NP’s stability.

In this revised manuscript, we have replaced the old stability measurements with the new and slightly numerically different values. We have also introduced a new figure supplement (Figure 6—figure supplement 3) that shows all CD traces. The melting curves (now Figure 6—figure supplement 4) are shown in non-normalized raw ellipticity, since normalization was disguising the problem of unequal concentrations in our earlier submission.

Below is a figure overlaying of all of the melting curves in non-normalized units. As can be seen from this figure (Figure 9), the samples in red had lower magnitude in the old measurement set, but now have magnitudes comparable to the rest of the data in the new measurement set.Author response image 1